

# Barriers to the use of personal health records by patients: a structured review

Chris Showell

eHealth Services Research Group, University of Tasmania, Australia

## ABSTRACT

**Introduction**. An increasing focus on personal electronic health records (PHRs) offers healthcare benefits for patients, particularly those in undeserved and marginalised populations, who are at risk of receiving less effective healthcare, and may have worse health outcomes. However, PHRs are likely to favour text, technical and health literate users, and be less suitable for disadvantaged patients. These concerns have prompted this review of the literature, which seeks evidence about barriers to the adoption and continued use of PHRs, the nature of the evidence for those barriers, and the stage of PHR implementation where particular barriers apply.

**Methods**. Searches in PubMed, Embase, CINAHL and ProQuest databases were used to retrieve articles published in English after 2003 in a refereed journal, or presented in a refereed conference or scientific meeting. After screening to remove items which were out of scope, the phase of the PHR implementation, the type of investigation, and PHR barriers were categorised using thematic coding.

**Results**. The search retrieved 395 items; screening identified 34 in-scope publications, which provided evidence of 21 identified barriers to patient adoption and continued use of PHRs, categorised here as Individual, Demographic, Capability, Health-related, PHR or Attitudinal factors. Barriers were identified in most phases of PHR implementation, and in most types of study. A secondary outcome identified that eleven of the publications may have introduced a bias by excluding participants who were less affluent, less capable, or marginalised.

**Conclusions**. PHR barriers can interfere with the decision to start using a PHR, with the adoption process, and with continued use, and the impact of particular barriers may vary at different phases of PHR adoption. The complex interrelationships which exist between many of the barriers is suggested in some publications, and emerges more clearly from this review. Many PHR barriers appear to be related to low socioeconomic status. A better understanding is needed of how the effect of barriers is manifested, how that effect can be countered, and how planning and implementation of PHR initiatives can make allowance for patient level barriers to PHR adoption and use, with appropriate actions to mitigate the effect of those barriers for more disadvantaged patients.

Corresponding author
Chris Showell, cmshowell@gmail.com

## INTRODUCTION

There is an increasing focus on personal electronic health records (PHRs) as a part of the implementation of ehealth services to support improvements in healthcare. PHRs have been defined as "...a private, secure application through which an individual may access, manage, and share his or her health information. The PHR can include information that

is entered by the consumer and/or data from other sources such as pharmacies, labs, and health care providers." (*Jones et al., 2010*, p. 334) Most publications about personal health record systems now focus on electronic versions which provide online access for patients, which may be through provider portals. PHRs offer a number of benefits including better access to data and information, improved communication between patients and providers, the empowerment of patients, and opportunities for health self-management (*Tang & Lansky, 2005*; *Pagliari, Detmer & Singleton, 2007*).

These benefits are certainly worthwhile, particularly for disadvantaged patients, who face challenges in receiving safe effective healthcare (*Adler & Newman, 2002*), and who are likely to have worse health outcomes than more privileged patients (*Olshansky et al., 2012*). However, the benefits which result from the use of a PHR cannot be guaranteed. The use of specialised medical language within a PHR can marginalise non-specialist users (*Showell, Cummings & Turner, 2010*), and in Australia, patients have largely been left out of discussions about policies affecting national PHR developments (*Showell, 2011*).

Information about demonstrated benefits to patients is limited. Most of the evidence of benefit applies to technically competent patients (*Green et al., 2008*; *Ralston et al., 2009*), with few details about how beneficial outcomes can be provided for other types of patients and patient groups. Concerns have been expressed previously about a risk that the development of PHRs may be skewed in favour of users with good levels of text, technical and health literacy; as a result PHRs may be less suitable for users who are at a socioeconomic disadvantage (*Showell & Turner, 2013a*; *Showell & Turner, 2013b*). Low levels of text, technical and health literacy can act as barriers to the effective use of technology (*Wilson, Wallin & Reiser, 2003*), including personal health records (*Angaran, 2011*; *Newman, Biedrzycki & Baum, 2012*), and a number of other barriers have been identified (*Sarkar et al., 2011*).

## OBJECTIVES

The concerns outlined above suggest that there are significant barriers to the adoption and continued use of PHRs by patients, particularly for those among disadvantaged and under-served populations. These barriers may relate specifically to the use of PHRs, or may entail more general problems with access to or the use of technology.

The broad intention of this literature review is to bring to the attention of informatics practitioners the range of issues and associated barriers which might prevent an equitable approach to PHR implementation.

The review is designed to address three specific questions:

- What patient level barriers to the adoption and continued use of PHRs have been identified?
- What is the nature of the evidence for each of those barriers?
- At what stage of PHR adoption and use are those barriers most likely to apply?

The review seeks information about those barriers, and the nature of the available evidence, as a way to establish, maintain and enhance equity in the development and implementation of PHRs. The intention is to provide an inclusive presentation of all identified barriers, and maintain the broadest possible scope.

## METHODS

### Eligibility criteria

The literature search identified publications providing evidence about barriers which might interfere with a patient's decision to adopt a personal health record, or discourage continued use. Publications were included if they considered any stage of patient involvement with a PHR, from their willingness or ability to use the internet or health information technology in the context of PHR use, through to long term use of a PHR as a part of their healthcare.

Publications in English after 2003, in a refereed journal, or presented in a refereed conference or scientific meeting were considered for inclusion. Publications were excluded if they focused on barriers affecting healthcare providers or organisations rather than patients, or if the description of barriers was not based on objective evidence, for example white papers, opinion pieces or editorials.

The types of publication which were sought included:

- Comparative trials involving multiple participating sites;
- Evaluations which involved the collection of data from patients about PHR barriers (using focus groups, interviews, surveys or questionnaires);
- Observational studies; and
- Details of the attitudes and opinions of patients about possible future PHR use.

The review considers the type of study reported, the number of participants in the study, and whether any aspects of the methodology in each case could make the identification of barriers less likely.

A conventional systematic review seeks to provide some degree of quantitative rigour within the findings. However, this structured review has applied a more inclusive, wide-ranging approach to the identification of barriers. Although raw counts of identified barriers are included in the text, there has been no attempt (or intention) to provide an overall qualitative assessment of barriers, or to evaluate their likely impact in particular settings.

### Study selection and data extraction

The review process followed published guidelines on Preferred Reporting Items for Systematic Reviews and Meta-Analyses (PRISMA) (*Liberati et al., 2009*). Full literature searches were conducted in PubMed, Embase, CINAHL and ProQuest databases between January and April 2014, with additional searches conducted in May 2014. Details were retrieved for all publications in English from January 2004 to the date of the search.

As an example, the search conducted in PubMed used the terms (personal health record OR personal electronic health records OR patient portal) AND (barrier OR barriers), retrieving 51 citations. Searches were also conducted in Embase, CINAHL and ProQuest using comparable search terms. Additional items were retrieved by tracking citations within publications, and from a small number of other sources.

All publications were initially screened to remove items which were considered to be out of scope, for example where the reference to PHRs was incidental (*Bonacina & Pinciroli, 2010*; *Abimbola et al., 2012*), where the barriers identified were exclusively those affecting healthcare providers and organisations (*Hart, 2009*; *Gaskin et al., 2011*), or where

the focus was on PHR infrastructure issues (*Hammond, 2005*; *Tejero & De la Torre, 2012*). The screening process also removed items which made only incidental mention of PHRs (*Stead, Kelly & Kolodner, 2005*) or barriers (*Burke et al., 2010*). Publications were included if they provided specific evidence about barriers which might influence the intended or actual adoption of PHRs by patients, or their continued use of a PHR.

Data from the publications which remained after screening were extracted using an iterative process of reviewing full text publications. The data variables which were recorded included the phase of PHR implementation, the type of investigation undertaken, barriers which were identified, the location of the study and the PHR system in use. Details were also recorded where relevant of the number of individuals in the population being studied, and the number included in the study. For studies which obtained information or participation from individuals, aspects of the methodology which might discourage or exclude low capability subjects from seeking to enrol in the study, or reduce the likelihood of their selection as participants were noted. Following an initial review of the data from all in-scope publications, frameworks were developed for the phase of PHR implementation studied, the type of investigation, and the evidence it provided about barriers.

### Implementation phase

For each publication, the authors' description of the phase of PHR implementation under investigation was reviewed, and thematic coding used to establish a schema describing each phase of implementation. This schema was then used to categorise all publications. The majority were focused on a single phase of implementation, with three (*Atreja et al., 2005*; *Cho et al., 2010*; *Luque et al., 2013*) addressing two phases.

### Investigation type

For each publication, descriptions of the type of study were reviewed, and used to develop a categorisation by type of investigation. Publications were assigned to a category of investigation type, with the majority of publications using a single type of investigation, and two (*Nijland et al., 2011*; *Gordon et al., 2012*) spanning two types.

### Barriers

Each of the publications was reviewed to identify evidence about barriers which might inhibit patients' adoption or continued use of a PHR, as well as barriers to internet use more generally (in the context of PHR use). An iterative process of thematic coding was used to classify barriers, with each included publication reviewed at least three times to ensure that meanings were not misinterpreted, and that the thematic structure remained consistent.

## RESULTS

### Summary

Searches in PubMed, Embase, CINAHL and ProQuest retrieved a total of 439 publications. Another 36 items were identified from citation tracking and other sources, giving a total of 475 publications. After removing 80 duplicates, 395 publications remained for initial screening. This resulted in the exclusion of 263 records, leaving 132 full text articles to be evaluated for eligibility. This evaluation removed 98 articles which provided no direct
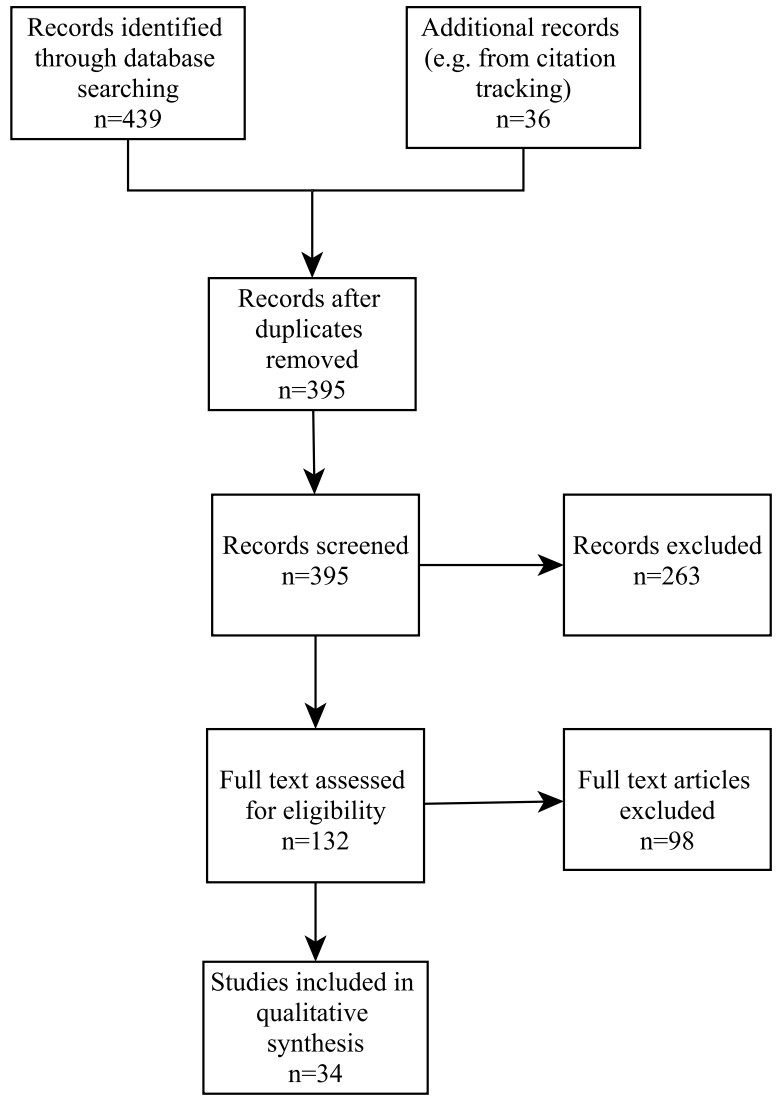

**Figure 1** PRISMA flowchart.

evidence about PHR barriers or did not address patient barriers to PHR adoption and use, and literature reviews. This left 34 articles for the synthesis of evidence. This process is outlined in Fig. 1.

Each of the included publications was coded in order to identify the particular phase of the PHR implementation which was being evaluated, the type of investigation conducted, and the barriers which were identified by the study.

## Key features of selected studies

Publications were categorised according to investigation type (data from users and non-users; observational studies; patient attitudes and opinions; or mixed). The four tables which follow are grouped by investigation type, and provide details of the included publications, including method, size of target population and number of participants.

**Table 1  Studies collecting data from PHR users or non-participants.**

| Author(s), year | Investigation type | Population | Participants |
|---|---|---|---|
| Anderson (2004) | Telephone interview survey | 3,000 | 186 |
| Atreja et al. (2005) | Focus groups/interviews with clinic staff; observation | – | 15 |
| Butler et al. (2013) | Telephone interviews with patients | 404 | 39 |
| Cho et al. (2010) | Postal survey questionnaire | – | 201 |
| Chrischilles et al. (2014) | Mixed methods: user-centred design with evaluation; questionnaire | 15,000 | 1,075 |
| Crabb, Rafie & Weingardt (2011) | Interview survey | 75 | 50 |
| Emani et al. (2012) | Postal survey questionnaire | 1,500 | 760 |
| Fuji, Abbott & Galt (2014) | Interviews with trained users | 59 | 23 |
| Goel et al. (2011a) | Telephone interviews with non adopters | – | 159 |
| Greenhalgh et al. (2008) | Mixed methods: Interviews/focus groups | – | 103/67 |
| Hall et al. (2014) | Trial of result communication via PHR | 66 | 49 |
| Hilton et al. (2012) | Online survey (within supported PHR use) | 2,871 | 338 |
| Kim et al. (2009) | Mixed methods: paper questionnaire, analysis of user logs | 330 | 70 |
| Kruse et al. (2012) | Interviews about internet use | 713 | 638 |
| Lober et al. (2006) | Analysis of data about PHR use | 170 | 41 |
| McCleary-Jones et al. (2013) | Interviews | 350 | 88 |
| Mishuris et al. (2014) | Semistructured interviews with patients | – | 3 |
| Roblin et al. (2009) | Paper survey with online option (non-adopters) | 5,309 | 1,777 |
| Taha et al. (2013) | Lab usability test of a simulated PHR | – | 107 |
| Weitzman, Kaci & Mandl (2009) | Focus groups, usability testing, email | – | 302 |

Table 1 summarises 20 studies involving the collection of data about barriers from PHR users, or participants who did not initiate or continue PHR use (using focus groups, interviews, surveys or questionnaires). One study in this category used semi structured interviews with health professionals about the characteristics of patients likely to use a PHR, as well as patient questionnaires. Barriers which were identified by both patients and clinic staff were included in the overall summation of barriers.

Table 2 outlines six observational studies which provide a qualitative or quantitative evaluation of demographic data and records of users and non-users, as well as patterns of activity for PHR users.

The six studies in the third category, which collected details of the attitudes and opinions of patients about barriers to possible future PHR use, and the demographic characteristics of those with particular usage intentions, are provided in Table 3.

Table 4 outlines two studies using more than one of the previous categories of investigation.

The publications retrieved for this review displayed a distinct geographic bias, with 32 out of a total of 34 studies reporting on PHR implementations in the USA (with one each from the United Kingdom and the Netherlands). Three particular PHR systems—MyHealtheVet, kp.org and MyChart—accounted for nine of the publications (with three each). Data about a possible bias in the selection of participants was retrieved during the data extraction, and evaluated as a secondary outcome.

**Table 2  Observational studies.**

| First author (yr) | Investigation type | Population | Participants |
| --- | --- | --- | --- |
| *Byczkowski, Munafo & Britto (2011)* | Retrospective observational study | 1,900 | 498 |
| *Goel et al. (2011b)* | Retrospective data analysis with adopters | 7,088 | 4,891 |
| *Nielsen, Halamka & Kinkel (2012)* | Retrospective chart review | 240 | 154 |
| *Sarkar et al. (2010)* | Telephone, web and written survey | 14,102 | 5,671 |
| *Sarkar et al. (2011)* | Telephone, web and written survey | 14,102 | 5,671 |
| *Yamin et al. (2011)* | Data analysis comparing adopters and non-adopters | 75,056 | 32,274 |

**Table 3  Patient attitudes and opinions.**

| First author (yr) | Investigation type | Population | Participants |
| --- | --- | --- | --- |
| *Logue & Effken (2012)* | Survey questionnaire | – | 38 |
| *Luque et al. (2013)* | Written questionnaire/Focus group | 120/8 | 90/- |
| *Noblin, Wan & Fottler (2012)* | Paper survey on health literacy and PHR usage intention | – | 562 |
| *Patel et al. (2011)* | Telephone survey | – | 200 |
| *Patel et al. (2012)* | Paper survey of support for Health Information Exchange and PHR | | 117 |
| *Zarcadoolas et al. (2013)* | Focus groups | – | 28 |

**Table 4  'Mixed methods' using multiple study types.**

| First author (yr) | Investigation type | Population | Participants |
| --- | --- | --- | --- |
| *Gordon et al. (2012)* | Mixed methods: surveys, database analysis, usage logs | 8,249 | 509 |
| *Nijland et al. (2011)* | Mixed methods: survey, interviews, log files, usability assessment | 350 | 50 |

## Implementation phase

For the purposes of categorising publications, the following schema was developed in order to identify which phase of PHR readiness, adoption and use was being studied in each investigation.

1. Readiness to use a PHR, including evaluations of internet use:
1.1—Patient use of technology, including the internet;
1.2—PHR usage intentions;
1.3—Design of PHRs with User Centred Design (UCD), or usability studies.
2. Initial registration for an account within a PHR system;
3. Initial use of a PHR; publications which studied any use of a PHR at an unspecified time after registration were included in this category;
4. Continued use of a PHR, including long term use;
5. PHR benefits affecting the patient's health and wellbeing.

## Barriers

Each of the included publications provided statements about barriers to patient adoption and continued use of a PHR. This evidence was either: described by users in advance or anticipation of PHR use; reported by potential users as a reason for not commencing use of

a PHR, or not continuing that use; or inferred from demographic differences between users and non-users. Thematic analysis was used to identify barriers and to categorise barrier types. This process involved a degree of simplification for some of the barriers described in publications.

Some concepts related to PHR barriers were unambiguous, and required little or no simplification. A reference to 'Age', for example, was taken as a straightforward description of a barrier, with no further interpretation required. However, some more diffuse concepts required a degree of interpretation. For example, "...problems due to reading, understanding and filling out forms, not due to poor vision" (*Sarkar et al., 2010*, p. e4) was recorded as a barrier resulting from poor health literacy, while an observation that "[p]articipants did not perceive the PHR as having added value for managing their existing self-care behaviors..." (*Fuji, Abbott & Galt, 2014*) was interpreted as a barrier related to 'Lack of Motivation'. The analysis identified 21 distinct barriers, which are listed by barrier category in Tables 5–10.

## Primary outcome: PHR barriers

Tables 5–10 summarise the barriers which were identified in each investigation type, and in each phase of PHR implementation for each barrier category. Most barriers were identified in most phases of PHR implementation, and in most types of study. Barriers which are likely to be associated with socioeconomic disadvantage are flagged. It should be noted that failure to identify a barrier within a particular publication does not provide evidence that the barrier was absent in the population studied, merely that it was not identified. It should also be noted that some of the publications report multiple phases or investigation types.

Each of the sections which follow provides additional information about a barrier category, and the barriers which were identified within that category.

## Individual characteristics

Barriers related to age, sex and race or ethnicity are innate characteristics of an individual user, not amenable to change, and were categorised as Individual factors.

### *Age*

A total of 13 of the included studies identified patient age as a barrier which has an impact on the adoption and continued use of PHRs. However, the effect was not clearly delineated. It is likely that age has a variable impact on ability, usage intention and motivation to continue using a PHR after enrolment. Internet use was more common for younger patients, with use declining with increasing age (*Kruse et al., 2012*). PHR 'innovators' were younger than other users and 'non-adopters' (*Emani et al., 2012*), with older patients less likely to enrol for a PHR (*Goel et al., 2011b*), although one study found that, once receiving a password, older patients were more likely to log on to the system (*Sarkar et al., 2011*).

### *Sex*

The sex of participants was noted as a barrier in statistical analyses, but the effect was generally modest, and inconsistent between publications. Studies found that men were more likely to find computer use enjoyable and be confident about using the internet and

**Table 5  Barriers related to individual characteristics.**

| | Age | Sex | Race/Ethnicity[*] |
|---|---|---|---|
| **Implementation phase:** | | | |
| 1 PHR readiness | | | |
| 1.1 Use of technology | 1 | | |
| 1.2 Usage intentions | 3 | 1 | |
| 1.3 Participation in design of PHRs | | 2 | |
| 2 Initial registration | 1 | 1 | 2 |
| 3 Initial use | 3 | | 2 |
| 4 Continued use | 2 | | 1 |
| 5 PHR benefits | | | |
| **Investigation type:** | | | |
| A Collection of data from PHR users, or non-participants | 8 | | 4 |
| B An observational study using demographic data and records of users and non-users | 3 | 2 | 4 |
| C Attitudes and opinions of patients about barriers | 2 | 1 | |

**Notes.**
*Barrier associated with socioeconomic disadvantage.

an online PHR (*Logue & Effken, 2012*), more likely to go online (*Cho et al., 2010*), and more likely to be higher users of PHRs, and more engaged (*Chrischilles et al., 2014*). However, one study (*Yamin et al., 2011*) found that women were 15% more likely to adopt a PHR (OR 1.15, CI [1.08–1.21]).

### Race and ethnicity

Race and ethnicity were identified as a barrier in eight studies, all undertaken in the USA. The studies found that racial and ethnic background could either inhibit the adoption of a PHR (*Kim et al., 2009*; *Roblin et al., 2009*; *Goel et al., 2011b*; *Emani et al., 2012*) or make its continued use less likely (*Yamin et al., 2011*; *Byczkowski, Munafo & Britto, 2011*; *Sarkar et al., 2011*). Publications did not always clarify the extent to which variations in PHR use were associated with related barriers such as education, income and socioeconomic status, literacy, or computer and internet access.

The predominant finding was that white patients were more likely to start and continue the use of a PHR, although one study (*Goel et al., 2011b*) found that while African-American and Hispanic patients were less likely to start using a PHR, their use of the system was no different once they were enrolled. Two studies (*Sarkar et al., 2011*; *Goel et al., 2011b*) found that adoption and use of a PHR was similar in white and Asian patients, while two others (*Yamin et al., 2011*; *Nielsen, Halamka & Kinkel, 2012*) found that use by Asian patients was less likely.

It should be noted that these specific findings in a US context may not be relevant in other countries, or with more recent arrivals.

## Demographic factors

Income, socioeconomic status, level of education, and internet and computer access were categorised as Demographic factors related to an individual's circumstances.

**Table 6  Barriers related to demographic factors.**

| | Income, socio-economic status[*] | Level of education[*] | Internet and computer access[*] |
|---|---|---|---|
| **Implementation phase:** | | | |
| 1 PHR readiness | | | |
|    1.1 Use of technology | 2 | 2 | 1 |
|    1.2 Usage intentions | 2 | 1 | 2 |
|    1.3 Participation in design of PHRs | | | |
| 2 Initial registration | | 1 | 3 |
| 3 Initial use | 2 | 1 | 2 |
| 4 Continued use | 2 | 1 | 2 |
| 5 PHR benefits | | | |
| **Investigation type:** | | | |
| A Collection of data from PHR users, or non-participants | 3 | 4 | 5 |
| B An observational study using demographic data and records of users and non-users | 3 | 1 | 1 |
| C Attitudes and opinions of patients about barriers | 2 | 1 | 2 |

**Notes.**
[*]Barrier associated with socioeconomic disadvantage.

### Income, socioeconomic status

PHR barriers for those with lower income and lower socioeconomic status were identified in eight studies. PHR adoption was less likely in groups with lower socioeconomic status (*Yamin et al., 2011*) and those without private health insurance, (*Byczkowski, Munafo & Britto, 2011*) although for those who did adopt a PHR, level of income did not appear to affect the degree of use (*Yamin et al., 2011*).

### Level of education

Level of education was identified as a barrier in six studies, associated with both computer and internet access and use (*Kruse et al., 2012*) and with the adoption and use of a PHR, (*Roblin et al., 2009*; *Emani et al., 2012*). The association between level of education and continued use of a PHR following enrolment appeared less pronounced (*Sarkar et al., 2011*).

### Internet and computer access

Lack of internet and lack of computer access were identified as barriers in ten studies. Problems with access did not appear to have a marked effect on PHR usage intention, (*Goel et al., 2011a*) although they did affect actual use of a PHR (*Lober et al., 2006*; *Nijland et al., 2011*; *Kruse et al., 2012*; *Luque et al., 2013*).

## Capabilities

Four barriers to PHR use were related to the skills and abilities of users and potential users. Functional or text literacy, numeracy, health literacy, and technical literacy and skills were assigned to the Capability factors category.

**Table 7   Barriers related to individuals' capabilities.**

|  | Text literacy/ functional literacy[*] | Numeracy[*] | Health literacy[*] | Technical literacy and skills[*] |
|---|---|---|---|---|
| **Implementation phase:** |  |  |  |  |
| 1 PHR readiness |  |  |  |  |
|    1.1 Use of technology |  |  | 1 | 1 |
|    1.2 Usage intentions |  |  | 2 | 3 |
|    1.3 Participation in design of PHRs | 1 |  |  | 2 |
| 2 Initial registration |  |  |  |  |
| 3 Initial use |  | 1 |  | 2 |
| 4 Continued use | 1 |  | 2 | 5 |
| 5 PHR benefits |  |  | 1 |  |
| **Investigation type:** |  |  |  |  |
| A Collection of data from PHR users, or non-participants | 1 | 1 | 4 | 7 |
| B An observational study using demographic data and records of users and non-users |  |  | 1 | 1 |
| C Attitudes and opinions of patients about barriers | 1 |  | 1 | 4 |

**Notes.**
  [*]Barrier associated with socioeconomic disadvantage.

### Text literacy/functional literacy

Only two studies specifically identified low levels of text literacy or functional literacy as a barrier to the use of a PHR, with functional literacy identified as a potential barrier by a focus group discussion (*Gordon et al., 2012*). This limited evidence was despite the obvious limitation that an inability to read would impose on a potential PHR user. The risk of introducing an unintended bias in a PHR evaluation by excluding subjects with poor literacy is considered in the Discussion section.

### Numeracy

Numeracy was identified as a barrier in only one study, with the authors finding that poor numeracy skills accounted for 4–5% of users' failures with overall task performance and the performance of complex tasks in a simulated PHR (*Taha et al., 2013*). It should be remembered, however, that an element of numeracy is often included as a contributor to overall health literacy.

### Health literacy

Low health literacy was identified as a barrier in six studies, and was noted as having an impact on both adoption (*Sarkar et al., 2011*; *Noblin, Wan & Fottler, 2012*) and continued use (*Lober et al., 2006*; *Kim et al., 2009*). Greenhalgh et al. *(2010)* found that many subjects who described their attitude to portal use as "...'not bothered' or 'don't care'..." were also judged by the researchers to have low levels of health literacy.

### Technical literacy and skills

Lack of technical literacy and lack of computer or internet skills were the most frequently identified barrier, with 13 publications identifying this as a barrier to either technology use

**Table 8  Health related barriers.**

| | Health, Chronic disease[*] | Disability (General)[*] | Physical disability[*] | Cognitive disability[*] | Visual disability[*] |
|---|---|---|---|---|---|
| **Implementation phase:** | | | | | |
| 1 PHR readiness | | | | | |
|    1.1 Use of technology | 1 | | | | 1 |
|    1.2 Usage intentions | 2 | | 1 | 1 | 1 |
|    1.3 Participation in design of PHRs | | | 1 | 1 | 1 |
| 2 Initial registration | 1 | | | | |
| 3 Initial use | 2 | 1 | | 1 | |
| 4 Continued use | 4 | | 1 | 2 | 1 |
| 5 PHR benefits | | | | | |
| **Investigation type:** | | | | | |
| A Collection of data from PHR users, or non-participants | 6 | | 2 | 4 | 3 |
| B An observational study using demographic data and records of users and non-users | 2 | 1 | | | |
| C Attitudes and opinions of patients about barriers | 2 | | | 1 | |

**Notes.**
[*]Barrier associated with socioeconomic disadvantage.

(*Adler & Newman, 2002*) or the use of a PHR (*Lober et al., 2006*; *Roblin et al., 2009*; *Nijland et al., 2011*; *Hilton et al., 2012*; *Butler et al., 2013*; *Luque et al., 2013*). Early adopters of a PHR were significantly more likely to self-report being 'comfortable' or 'very comfortable' with internet use (*Butler et al., 2013*) while those with rudimentary computer skills showed little improvement in PHR use over time (*Hilton et al., 2012*).

## Health related

Barriers resulting from the individual's health and wellbeing, including the presence of a chronic disease, disability generally, and specific physical, cognitive or visual limitations, were categorised as Health related.

### Health, chronic disease

Data from ten studies identified a complex relationship between health and both internet use and PHR adoption and use. Those whose self-reported health status was excellent or very good were more likely to be internet users (*Kruse et al., 2012*), while patients with poorer health overall were less likely to adopt a PHR (*Emani et al., 2012*). However, those with multiple comorbidities were identified as being more likely to adopt a PHR (*Roblin et al., 2009*; *Emani et al., 2012*) or expressed willingness to choose a healthcare provider based on the provider's use of information from their PHR (*Logue & Effken, 2012*).

### Disability

Disability can create practical barriers to the use of information technology, including PHRs (*Angaran, 2011*). One publication identified disability as a generic barrier to PHR use; physical impairment was identified in two studies; cognitive impairment in five studies; and visual impairment in three studies. Physical, visual and cognitive impairment have all been identified as barriers to successful use of a PHR (*Lober et al., 2006*; *Kim*
**Table 9  Barriers related to the PHR itself.**

| | Usability | Cost | Lack of information |
|---|---|---|---|
| **Implementation phase:** | | | |
| 1 PHR readiness | | | |
|    1.1 Use of technology | | 1 | |
|    1.2 Usage intentions | 1 | | |
|    1.3 Participation in design of PHRs | 1 | 1 | |
| 2 Initial registration | | | 2 |
| 3 Initial use | | | |
| 4 Continued use | 1 | | |
| 5 PHR benefits | | | |
| **Investigation type:** | | | |
| A Collection of data from PHR users, or non-participants | 2 | 1 | 2 |
| B An observational study using demographic data and records of users and non-users | | | |
| C Attitudes and opinions of patients about barriers | 1 | 1 | |

*et al., 2009*), although design adaptations may help to reduce the severity of those barriers (*Atreja et al., 2005*).

## PHR factors

Barriers associated with the usability of a PHR, the costs associated with access, or lack of information about the PHR were categorised as PHR factors.

### Usability

Three studies identified usability as a barrier to successful adoption and use of a PHR by patients. One study which looked for specific barriers affecting patients with multiple sclerosis (*Atreja et al., 2005*) found that issues such as a cluttered display, small font size, and poor contrast created barriers, while another (*Fuji, Abbott & Galt, 2014*) reported patient difficulties with navigation between pages, and the need for repeated clicking during data entry.

### Cost

Two studies identified costs to users as a barrier for PHRs, with patients reporting that they could not afford the cost of a computer and a broadband internet connection (*Kruse et al., 2012*; *Luque et al., 2013*).

### Lack of information

Two studies identified that a lack of information about the availability of a particular PHR (*Mishuris et al., 2014*), or accessibility of information about options within a PHR (*Atreja et al., 2005*) could interfere with use.

## Attitudinal factors

The remaining barriers—discomfort with computer use, concerns about privacy, security and confidentiality, and lack of motivation—were categorised as Attitudinal factors.

**Table 10  Barriers related to individuals' attitudes to PHRs.**

|  | Discomfort with computer use | Privacy and confidentiality concerns | Lack of motivation |
|---|---|---|---|
| **Implementation phase:** | | | |
| 1 PHR readiness | | | |
| 1.1 Use of technology | 1 | 2 | |
| 1.2 Usage intentions. | | 2 | 1 |
| 1.3 Participation in design of PHRs, | 2 | 2 | |
| 2 Initial registration | | 1 | 2 |
| 3 Initial use | 1 | | |
| 4 Continued use | 2 | | 1 |
| 5 PHR benefits | | | |
| **Investigation type:** | | | |
| A Collection of data from PHR users, or non-participants | 4 | 3 | 2 |
| B An observational study using demographic data and records of users and non-users; | | | 1 |
| C Attitudes and opinions of patients about barriers | | 4 | 1 |

### *Discomfort with computer use*

Four studies identified some form of discomfort with the use of a computer (*Kruse et al., 2012*) as a barrier to the adoption and use of a PHR. This barrier was also described as a lack of confidence and fear of failure, and as 'computer anxiety' (*Lober et al., 2006*; *Kim et al., 2009*).

### *Privacy and confidentiality concerns*

Patient concerns about privacy, security or confidentiality of the personal health information stored in a PHR were reported in seven studies (*Anderson, 2004*; *McCleary-Jones et al., 2013*). In some cases these concerns were specifically related to the need to access a PHR from a public or shared computer (*Luque et al., 2013*; *Mishuris et al., 2014*).

### *Lack of motivation*

Three studies provided evidence that a lack of motivation could be a barrier to the use of a PHR. Potential users did not see the PHR as providing added value (*Fuji, Abbott & Galt, 2014*; *Mishuris et al., 2014*) or thought that using a PHR would take up too much time (*Nijland et al., 2011*; *Fuji, Abbott & Galt, 2014*).

## SECONDARY OUTCOME: SELECTION BIAS

Eleven of the publications which identified PHR barriers introduced a potential bias by using a data collection methodology which could exclude participants who were less affluent, less capable, or marginalised. Those methodological choices fell into four broad categories, with one publication (*McCleary-Jones et al., 2013*) including more than one type of bias:

### A focus on those already using technology

In five publications participation was restricted to subjects who already had experience using a web browser (*Lin et al., 2005*; *Nijland et al., 2011*), had an existing portal account (*Byczkowski, Munafo & Britto, 2011*) who had received training in the use of a PHR (*Fuji, Abbott & Galt, 2014*), or who were required to complete web based surveys during the study (*Hilton et al., 2012*). These studies did not report barriers related to Capability factors, or to disability.

### Exclusion of participants with serious illness or infirmity

In two publications subjects were excluded if they were prevented from participating in an interview as a result of a serious comorbidity (*Atreja et al., 2005*) or if obvious cognitive deficits were observed (*McCleary-Jones et al., 2013*). These studies did not report any barriers associated with Individual or Demographic factors, and only health literacy was identified as a Capability factor.

### Excluding participants on the basis of language and literacy

Selection of participants for four of the studies (*Logue & Effken, 2012*; *Kruse et al., 2012*; *Patel et al., 2012*; *McCleary-Jones et al., 2013*) required them to be able to speak, read or write English. These studies identified a wide range of barriers in all categories (11 in all).

### Selection of subjects from within a population less likely to be disadvantaged

In these three publications data collection was restricted to participants with a landline telephone (*Anderson, 2004*), to university undergraduates in schools of business and information systems (*Whetstone & Goldsmith, 2009*), or to members of a community less likely to be disadvantaged (*McCleary-Jones et al., 2013*). Health literacy and privacy concerns were the only barriers to PHR adoption and use which were identified in these studies.

Identification of these potential sources of bias is not intended as a criticism of the studies, or of the authors. However, inadvertent bias within the methodologies of studies may mean that any evaluation of barriers within publications (such as that provided by this review) is likely to underestimate the prevalence and significance of barriers, particularly if those barriers are related to exclusion criteria which have been applied in the selection of participants.

## DISCUSSION

### Barriers

This literature review has identified evidence for 21 barriers, categorised as Individual, Demographic, Capability, Health related, PHR related and Attitudinal factors, which could interfere with or prevent a patient's adoption or continued use of a personal health record. The evidence is consistent, with ten of the barriers being identified in six or more publications. However, the frequency with which a particular barrier is identified provides little indication of that barrier's overall significance, or of its importance in particular settings. The low incidence (four publications or fewer) of reports identifying

text literacy, numeracy, generalised disability, or physical and visual impairment as barriers is more likely to result from the research methodology and from the relative invisibility of disadvantaged participants, rather than from the insignificance or absence of these barriers. The complex interrelationship which exists between many of the barriers is suggested in some publications, and emerges more clearly from this review. Socioeconomic status and educational attainment are closely related, and associated with text, technical and health literacy, and with numeracy; internet and computer access, computer skills and discomfort with the use of a computer are closely intertwined; and lastly PHR usability is likely to have a greater impact on users with lower capabilities, and PHR costs will be more challenging for poorer patients. Furthermore, socioeconomic disadvantage is likely to be statistically more prevalent among older citizens, and within non-Caucasian communities. The review identified predominantly US studies, which identify specific issues for elderly, African-American, and Latino communities.

### Barriers by type of investigation

The evidence about barriers to PHR adoption and use varies with the types of investigation. Firstly, data collected from patients themselves provides direct evidence about actual barriers which they face in adopting and continuing to use a PHR, although there may be a tendency for self-reports to underestimate the importance of barriers such as socioeconomic status, text literacy, health literacy and numeracy, all of which can carry a social stigma. Secondly, observational studies using PHR usage logs and health administrative data for PHR users and non-users can provide evidence about barriers, but only from an analysis of the data items which are included in those records. In many cases socioeconomic status, text and health literacy, or computer and internet use are not recorded, although an area measure of socioeconomic status can be imputed from the patient's home address. Finally, attitudes and opinions of patients about PHR benefits and barriers, and usage intention can be instructive, although there may be a gap between stated intention and future actions.

The 'Diffusion of Innovations' theory (*Rogers, 1983*) which is sometimes applied to the uptake of systems such as a PHR embodies an assumption that all potential users will eventually begin using a new system. The 'Technology Acceptance Model' (*Davis, 1989*) provides a more pragmatic approach, and suggests that actual system use is driven by an individual's perception about ease of use and usefulness, and by his or her attitudes and behavioural intentions to the system. However, initial perceptions about usefulness and ease of use may not be matched by the reality of the system itself. This perception-reality gap may be greater among potential users who have little or no previous experience with such systems, and initial attempts to use a new system may not translate into continued long-term use.

### Barriers by phase of implementation

Evidence about PHR barriers also varies by the phase of implementation being investigated. In Phases 1 and 2 (pre-adoption and initial registration) evidence about barriers is most likely to be about usage intention. Evidence suggests a gap between usage intention and actual PHR use. Disadvantaged and low capability users may see use of a PHR as beneficial, but may overestimate their own capabilities, and underestimate the demands

and challenges involved in using a PHR. Individuals may lack full awareness of the extent of their limitations, or may not see those limitations as making PHR use more difficult. In Phase 3 (early use) enthusiasm about first use may revert to a lack of interest once the effort required to use a PHR becomes apparent; evidence about barriers from evaluations of registration and first use are likely to provide an indication of those barriers which might interfere with the decision to use a PHR, while barriers identified in Phase 4 (continued use) provide insight into the constraints which are likely to interfere with long term use. Depending on the particular PHR, maintaining regular use could be difficult, although moderated by the skills and capabilities of the user. A continued interest by patients in using a PHR is likely to be influenced by perceptions of healthcare needs, and how those needs are met by a PHR, relative to other care that they receive. Barriers may also be context-sensitive, and influenced by PHR usability and user capabilities. PHRs need to be suitable for all users; testing with volunteers with good text, technical and health literacy may overestimate the suitability of the PHR for a broader population.

## Bias

A number of the included studies chose participants in a way that might result in a lower proportion of disadvantaged and low-capability users, compared with the overall population, resulting in a probable underestimate of PHR barriers. Some degree of bias may be unavoidable. Acquiring evidence about PHRs, including evidence about barriers, must rely on subjects who are able to participate: studies of PHR usage must rely on PHR users, participants must read a written questionnaire in order to respond, and it can be difficult to ethically engage research subjects with cognitive limitations. On the other hand, PHRs are intended for users who are unwell, not just healthy, educated, well-off patients. One study (*Zarcadoolas et al., 2013*) (not included in the evaluation of bias) deliberately introduced an inverse bias by seeking out participants with a low socioeconomic status.

### Limitations

This review has produced a biased evaluation of PHR barriers. Selecting publications in English has given an Anglophone, US-centric account of PHR barriers, from a restricted range of study sites, with little information from other countries. There may also be a publication bias: many of the publications from the USA are from large (and possibly well resourced) healthcare organisations and academic institutions able to provide early support for PHR users; results for PHR implementations in smaller, less well resourced settings might report barriers differently.

## CONCLUSIONS

### Principal findings

This review has found evidence of a range of barriers which interfere with the adoption and continued use of PHRs, with 111 instances of 21 distinct barriers identified across 34 publications. This evidence was found in all types of investigation, and in all phases of PHR adoption. Further research may find other as yet unidentified barriers, as well as variants of barriers identified in this review. A close relationship is evident between

socioeconomic status and PHR barriers, with 13 of the 21 barriers being associated with socioeconomic disadvantage. This confirms that the use of a PHR is likely to be harder for disadvantaged patients; PHRs as they are currently implemented may not provide a universal solution for problems with healthcare delivery or communication. The relative importance of a PHR barrier cannot easily be deduced from the number of times that it appears within the research literature. Rather, there is an obligation during PHR design, and during PHR implementation, to make a careful assessment of the likelihood of each barrier being present within the population being considered as users. In the USA, the Meaningful Use Stage 2 compliance criterion for 2017, which requires that 5% of patients access their record (*Centers for Medicare and Medicaid Services, 2017*), is more likely to measure record access by competent PHR 'early adopters' than by disadvantaged users. Despite the problem of a growing 'ehealth divide' (*Cummings, Chau & Turner, 2008*) this criterion as currently defined provides little impetus for health professionals or hospitals to encourage PHR enrolments among disadvantaged patients.

## Future research priorities

While this review has identified a broad range of PHR barriers, there was insufficient consistency across multiple studies to provide a comprehensive picture of the effect of barriers during PHR implementation and use. If those barriers affecting the population of potential users are to be addressed early in the process of design and implementation, there is a need for better identification and characterisation of both barriers and users. As Kushniruk and Turner have observed, "...greater consideration of who the user is and how the user is involved and their inputs mediated needs to be further articulated. To address these issues it is useful to try to be more precise about who the users are, when and where they are engaged, what expectations we have about our users and why." (*Kushniruk & Turner, 2011*, p. 281). Developing a better understanding of the impact of barriers on PHR users will help to ensure more effective use of the resources allocated to PHR systems. There is also a need for a better appreciation of how barriers can affect PHR adoption and use, and how that effect can be countered. Simply being aware of the possibility that a particular barrier may inhibit PHR use for some patients should be enough to ensure that this barrier is taken into account during PHR design and implementation. However, the apparent bias evident in a number of the studies suggests that the existence and significance of barriers is not universally recognised, and that further research may be warranted in order to provide stronger evidence. Finally, the results of this literature review raise a number of interesting questions which may suggest possibilities for future research:

- What does a PHR designed specifically for 'low functional literacy' users look like?
- What assistive options within a PHR could help to reduce the negative impact of poor health literacy?
- How can attention to PHR design minimise the impact of cognitive limitations for older patients?

The response to these questions may help to identify a path towards PHRs designed for specific groups of disadvantaged patients, or with an interface which is sufficiently simple, and adaptable to meet the needs of all users.

## ACKNOWLEDGEMENTS

My thanks go to Associate Professor Paul Turner and Dr Liz Cummings from the University of Tasmania, who provided unstinting encouragement, advice and support during the preparation of this review.

### Funding

The author received no funding for this work.

### Competing Interests

The author declares there are no competing interests.

### Author Contributions

- Chris Showell conceived and designed the experiments, performed the experiments, analyzed the data, contributed reagents/materials/analysis tools, wrote the paper, prepared figures and/or tables, reviewed drafts of the paper.

### Data Availability

The raw data is included in the tables in the manuscript.

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
