# Peer review of "Barriers to the use of personal health records by patients: a structured review"

_PeerJ, doi:10.7717/peerj.3268_

## Round 0.1 · original submission · Major Revisions

Thank you for your submission. The reviewers note that there are several areas that should be addressed. Particular attention should be paid to the comments relating to experimental design and the validity of your findings.

Reviewer 1 ·

Basic reporting

1) The authors should double check their References list for completeness. For example, the Archer, 2011 reference is not included.

Experimental design

1) The eligibility criteria for the literature review was very broad and goes beyond PHR use to include Internet use generally. While PHR use and Internet use are obviously linked, this linkage and the associated barriers are intuitive (e.g. no computer access limits PHR use; no Internet access limits PHR use). The authors may be better served by focusing on more specific barriers for PHR use.

2) As I understand the methodology for literature reviews, reviews themselves should not be included, and instead, the individual studies that compose the review should be extracted and included. This is to avoid the potential for duplication of findings. While the authors did note that “Literature reviews were removed if they included only publications which had already been retrieved for this review…” this does not completely eliminate concerns with duplication of findings.

3) The authors should discuss the limitations of including literature with provider perceptions of patient barriers, considering that providers are not the primary end users of these tools (although they certainly play an important role in supporting its use).

4) The authors briefly discuss the limitations of including potential users, but should explicate this limitation more clearly (i.e. potential use or perceived projected use is not necessarily a good indicator of future use).

Validity of the findings

1) It is not clear to me how the category barrier of “health literacy” could include statements such as “lack of motivation” which may have nothing to do with issues of health literacy and could have more to do with integration of these types of tools into a daily routine or even usability issues. Similarly, lack of added value sounds more like a functionality issue than a health literacy issue.

2) Why is education not flagged as a barrier likely to be associated with socioeconomic disadvantage?

3) Under 3.7.3 Race and ethnicity – the authors need to provide information about which races/ethnicities were less likely to adopt a PHR.

4) In the discussion of limitations, the authors note that a limitation was studies that focused on those already using technology may underestimate the prevalence and significance of barriers. While this is certainly true, the authors should also consider that barriers encountered by those who are already using technology may be uncovering particularly large issues, as some of those barriers identified may have been unexpected in this patient population.

Additional comments

1) In Table 2, there are no totals provided for “Health related factors” and nothing filled out at all for “PHR factors” and “Attitudinal factors”.

·

Basic reporting

The subject is interesting and relevant to modern medicine
Basic reporting is good
Figures are relevant and clear
Appropriate data is made available
Sections should have the numerical prefixes removed - "Eligibility criteria" instead of "2.1 Eligibility criteria"

Comments on tables -
Table 1 - UCD not defined for Chrischilles 2014 article

Close parentheses for Hilton (2012) Online survey (within supported
PHR use AND Logue (2012) Survey questionnaire)

HIE not defined for Patel (2012)

Consider condensing or splitting the table so it is no larger than page sized - perhaps a table for each major study type.
* * *
Table 2 is poorly designed and should be reformatted.

Consider a table for each factor type - capability, health related factors with a row for each of the items currently in the columns. Instead of item 1.1 use "PHR readiness - use of technology" and compare that method to the factor types

Something like -
Useability Cost
PHR readiness - use of technology
PHR readiness - usage intentions.
PHR readiness - participation in design of PHRs,

Experimental design

Experimental design is appropriate and adheres to the PRISMA format
The research question is appropriately selected and addressed by the study

Validity of the findings

The data is sound and the conclusions drawn from the data are appropriate.

---

## Round 0.2 · accepted · Accept

Thank you for your submission, we have now accepted your article for publication

·

Basic reporting

This paper is well written and referenced. It discusses a highly relevant topic for healthcare and explores the reported barriers to the adoption of a technology thought to be central to a patient centered healthcare delivery system.

1. Basic Reporting
Clear and unambiguous, professional English used throughout.
Yes

Literature references, sufficient field background/context provided.
Yes

Professional article structure, figs, tables. Raw data shared.
Yes. Revised tables are excellent.

Figures should be relevant to the content of the article, of sufficient resolution,
Yes

Self-contained with relevant results to hypotheses.
Yes

Experimental design

2. Experimental design
Original primary research within Aims and Scope of the journal.
Yes

The submission should clearly define the research question, which must be relevant and meaningful. The knowledge gap being investigated should be identified, and statements should be made as to how the study contributes to filling that gap.
Yes

Rigorous investigation performed to a high technical & ethical standard.

Yes. The authors changes represent significant improvements to the article that clarify areas identified previously. The included data on provider perceptions of barriers is useful because the attitude of providers to the PHR is likely to be an important factor driving end user acceptance.

Methods described with sufficient detail & information to replicate.
Yes

Validity of the findings

3. Validity of the Findings
Impact and novelty not assessed. Negative/inconclusive results accepted. Meaningful replication encouraged where rationale & benefit to literature is clearly stated.
Yes


Data is robust, statistically sound, & controlled.
Yes


Conclusion are well stated, linked to original research question & limited to supporting results.
Yes